# Long Noncoding RNAs: Recent Insights into Their Role in Male Infertility and Their Potential as Biomarkers and Therapeutic Targets

**DOI:** 10.3390/ijms222413579

**Published:** 2021-12-18

**Authors:** Shanjiang Zhao, Nuo Heng, Bahlibi Weldegebriall Sahlu, Huan Wang, Huabin Zhu

**Affiliations:** 1Embryo Biotechnology and Reproduction Laboratory, Institute of Animal Science, Chinese Academy of Agricultural Sciences, Beijing 100193, China; zhaoshanjiang@caas.cn (S.Z.); hengnuobj@foxmail.com (N.H.); blenbah@gmail.com (B.W.S.); 17710734315@163.com (H.W.); 2Tigray Agricultural Research Institute, Mekelle Agricultural Research Center, Mekelle 7000, Ethiopia

**Keywords:** long noncoding RNAs, male infertility, spermatogonia proliferation, spermatogonia differentiation, spermatocyte meiosis, potential therapeutic targets

## Abstract

Long noncoding RNAs (lncRNAs) are composed of nucleotides located in the nucleus and cytoplasm; these are transcribed by RNA polymerase II and are greater than 200 nt in length. LncRNAs fulfill important functions in a variety of biological processes, including genome imprinting, cell differentiation, apoptosis, stem cell pluripotency, X chromosome inactivation and nuclear transport. As high throughput sequencing technology develops, a substantial number of lncRNAs have been found to be related to a variety of biological processes, such as development of the testes, maintaining the self-renewal and differentiation of spermatogonial stem cells, and regulating spermatocyte meiosis. These indicate that lncRNAs can be used as biomarkers and potential therapeutic targets for male infertility. However, only a few comprehensive reviews have described the role of lncRNAs in male reproduction. In this paper, we summarize recent findings relating to the role of lncRNAs in spermatogenesis, their potential as biomarkers for male infertility and the relationship between reproductive arrest and transgenerational effects. Finally, we suggest specific targets for the treatment of male infertility from the perspective of lncRNAs.

## 1. Introduction

Only 4% of the human genome features RNA that is translated into protein [1]. The remaining RNAs are designated as noncoding RNAs (ncRNAs) and are considered to be “transcriptional noise” [2,3]. Although ncRNAs do not contain an open reading frame (OFR) and cannot transcribe protein, they are involved in the regulation of gene expression [4,5], biological development and metabolic diseases [6,7]. Studies have also shown that ncRNAs act as epigenetic regulators [8] and riboregulators [9], which play vital roles in the early stages and post-transcriptional genetic control of spermatogenesis and oogenesis. These indicate that they can be used as biomarkers for reproductive processes and diseases [10,11,12]. Most of the ncRNAs involved in mammalian reproduction are small ncRNAs (sncRNAs) [13,14,15,16]. As research effort intensifies in the reproductive sciences, many long noncoding RNAs (lncRNAs) have been shown to be involved in a variety of male reproductive processes, such as the development of testes [17], maintenance of spermatogonial stem cells (SSCs) self-renewal [18,19], and spermatogenesis [20]. Furthermore, many lncRNAs have been confirmed as potential biomarkers for male infertility [21,22,23].

LncRNAs are composed of nucleotides located in the nucleus and cytoplasm, which are transcribed by RNA polymerase II and greater than 200 nt in length. The level of transcription for genes transcribing lncRNAs is lower than those of protein-coding genes [24,25,26]. LncRNAs are known to perform multiple functions in mammals and regulate gene expression at several levels [27,28]. For example, in the pre-transcription stage, lncRNAs regulate histone modification and DNA methylation [29]. During the transcription stage, lncRNAs regulate some enhancer activity [30] and transcription factors [31]. Moreover, lncRNAs also control the expression of the RNA subcellular structure, RNA stability and post-transcriptional events [32]. LncRNAs are known to execute multiple functions in mammals; this is because (1) they can easily combine with homologous DNA sequences (transcribed lncRNA gene sequences and genes with similar sequences); (2) they can combine with homologous RNA with similar sequences; and (3) they can fold to form complex secondary structures and bind to a variety of proteins.

With the development of high throughput sequencing technology, many functional lncRNAs have been discovered, and many of these lncRNAs are thought to be involved in the formation of spermatozoa [33,34,35,36]. The expression profiles of lncRNAs in germ cells at different developmental stages were identified in mouse testes using gene chips and a large number of differentially expressed lncRNAs (DELs) were detected at different developmental stages [37]. This previous study showed that the expression levels of more than 900 lncRNAs were up-regulated in germ cells during mouse spermatogenesis and that different lncRNAs appeared during specific developmental stages [37]. Another study compared the testicular tissues of both newborn and adult mice and detected 3025 DELs in testicular lncRNA expression profiles [38]. The differential expression of lncRNAs at different developmental stages in the testicular tissue may be related to the process of spermatogenesis. In large livestock, Wang et al. [39] reported that lncRNAs are important in the regulation of bull sperm motility, and a total of 2517 lncRNAs were differentially expressed when compared between Holstein bulls with low or high sperm motility. In stallion sperm, 11,215 lncRNAs were identified, and 166 DELs were found to differ between groups of stallions living at greater and lower densities [40]. Because associations have been reported between many lncRNAs and reproductive disorders, it is very prudent to characterize and identify lncRNAs by sequencing and bioinformatic approaches as these may help us to determine the causes of male infertility [41]. However, few authors have attempted to review the role of lncRNAs in male reproduction, particularly with regard to their potential as therapeutic biomarkers. In this paper, we discuss recent findings relating to the role of lncRNAs in the SSCs, meiosis and spermatogenesis. Furthermore, we review recent insight into the role of lncRNAs in male reproductive arrest and their potential as therapeutic targets for the treatment of male infertility. This review provides a deeper insight into male infertility and discusses the biological mechanisms involved and their classification.

## 2. Classification and Regulatory Mechanisms Underlying the Action of Long Noncoding RNAs

### 2.1. The Classification of Long Noncoding RNAs

LncRNAs do not directly encode for proteins. However, an increasing number of studies have rejected their classification as “junk” and confirmed the important biological functions that they play in a variety of biological processes, including genome imprinting [42], cell differentiation [43], apoptosis [44], stem cell pluripotency [45], X chromosome inactivation [46], and nuclear transport [47,48,49]. The biological functions of lncRNAs are described in detail in our previous paper [20], which mainly focused on classification and regulatory mechanisms. LncRNAs can be classified according to their locations in relation to neighboring protein-coding genes and their mechanisms of action [41,50,51]. According to their location, lncRNAs can be classified as (1) intronic lncRNAs which reside within introns and are transcribed in the intronic region of protein-coding genes; (2) intergenic lncRNAs, also referred to as lincRNAs, situated between protein-coding genes and located at least 1kb away from the nearest protein; (3) stand-alone lncRNAs which are transcribed at different genomic sites and do not overlap with protein-coding genes; (4) antisense lncRNAs which are usually antisense strand transcripts of protein-coding genes and have sequence overlap with the mRNA of that gene. About 70% of genes have antisense lncRNAs; (5) enhancer lncRNAs which may be transcribed from enhancers via RNA poll II; and (6) promoter lncRNAs which are transcribed from promoters via RNA poll II [52]. LncRNAs are transcribed in opposite directions in the protein-coding gene, with transcription starting at the 3′ end of the protein-coding gene, and overlap one or more of the coding exons (Figure 1).

According to their mechanism of action, lncRNAs can be classified into four types (Figure 2): signal, decoy, guide, and scaffold [53,54]. The signal mechanism involves lncRNAs regulating downstream genes [55]. Previous studies have shown that lncRNAs can regulate the expression of downstream genes after lncRNA transcription [55,56]. LncRNAs are transcribed specifically in different signal pathways and participate in the action of specific signal pathways as signal transduction molecules. The regulatory process used by lncRNAs for gene transcription is very rapid because the signaling pathways involved are not involved in the translation of proteins [57]. In the decoy mechanism, lncRNAs block molecular function and mediate downstream signaling pathways [58]. Once an lncRNA has been transcribed, it can directly combine with a transcription factor or transcription regulator protein. The combination of lncRNA with a transcription factor or a regulator can block protein functions and regulate the transcription of downstream genes [59,60]. A further lncRNA decoy mechanism is the competing endogenous RNA (ceRNA) mechanism [61,62]. MicroRNA (miRNA) can typically bind with partial complementarity to sequences in the 3′ untranslated (UTR) regions of target mRNAs, and then induce mRNA degradation [62]. However, some lncRNAs and mRNAs show high degrees of sequence homology. Therefore, miRNA could bind to the 3′UTR of a target gene through sequence complementarity, and bind to lncRNAs with high sequence similarity. Where the amount of miRNA in a cell is limited, competitive binding between the lncRNAs and mRNA leads to a ceRNA mechanism [63,64]. With regard to the guide mechanism, lncRNAs can combine with transcription factors or regulators to form compounds that are located on specific DNA sequences that are involved in transcriptional regulation [50]. A previous study identified an additional lncRNA scaffold mechanism in which multiple related transcription factors were able to bind to the same lncRNA [65,66]. When multiple signal pathways are activated, the downstream effectors of different signal pathways are able to bind to the same lncRNA molecule to integrate information between different pathways. This scaffold mechanism enables cells to quickly respond and regulate a particular stimulus [67,68]. 

### 2.2. The Regulatory Mechanisms of Long Noncoding RNAs

Chromatin has two states: the euchromatin state, which is open and can be transcribed, and the heterochromatin state, which is a compact DNA-protein structure that cannot be transcribed [69]. Pre-transcriptional regulation refers to the regulatory events that occur in the preparatory phase prior to a transcriptional event and involves the states of chromatin [70]. Once the transition from heterochromatin to euchromatin is complete, a gene can undergo transcription and vice versa (Figure 3) [70]. The state of chromatin is determined by epistatic modifiers: the euchromatin state is enriched with activating histone modifications, such as H3K4me3, H3K36me3 and histone acetylation [71]. While the heterochromatin state is enriched with inhibitory histone modifications such as H3K9me3, H3K27me3, H4K20me3 and DNA methylation [72]. Interestingly, lncRNAs, the transcripts that are transcribed from the partial sequences of its target genes, have been shown to be synergistically involved in the precise regulation of the chromatin state by epistatic modifiers [73]. During this stage, part of the single-stranded region of the lncRNA binds to the target homologous DNA sequences (this allows transcription of the lncRNA gene sequences and genes with similar sequences). Next, other parts of the region fold into higher structures that bind to epistatic factors and help to regulate the conversion of the histone modification state. This, in turn, regulates the euchromatin or heterochromatin of the chromatin state.

During transcription, RNA poll II binds to the core promoter via transcription factors. DNA then undergoes local deconvolution and transcription is suspended due to the low level of phosphorylation at the C terminal domain (CTD) of RNA poll II. At this stage, lncRNAs are able to inhibit the binding of the RNA poll II complex to the promoter to complete transcriptional interference [74,75]. During transcriptional elongation, transcription factors (TFs) can bind to enhancer sequences cause high phosphorylation of RNA poll II, thereby accelerating transcriptional elongation. LncRNAs can regulate transcriptional elongation by capturing TFs, and then regulate enhancer activity, resulting in reduced levels of RNA poly II phosphorylation [76].

Post-transcriptional regulation is mainly concerned with the processing, translocation, and stability of the mRNA obtained by transcription [77,78]. During this stage, lncRNAs take advantage of their ability to bind easily to RNA to perform regulatory roles. During the splicing of mRNA, if the lncRNA binds to the splice site, it will then result in the site not being recognized by the splicing factor, preventing splicing [79]. During mRNA transport, some sequences of nuclear-localized lncRNA can bind to similar sequences of mRNA, thus resulting in the failure of mRNA transport [80]. LncRNAs can directly regulate RNA stability by binding to RNA targets and forming structural domains that are easily degraded or become more robust [81]. Furthermore, microRNA response elements are present on mRNAs, thus, miRNAs can bind to mRNAs and inhibit mRNA translation or cause mRNA degradation. However, lncRNAs can bind to miRNAs to form ceRNA regulatory mechanisms, thus indirectly regulating mRNA translation or degradation.

## 3. The Role of LncRNAs in Spermatogenesis

Spermatogenesis is an intricate developmental process in which SSCs differentiate from gonadal cells to form type A spermatogonia, type B spermatogonia, primary spermatogonia, secondary spermatogonia, and finally spermatozoa [20,82,83]. During this process, genes relating to spermatogenesis are expressed in highly specific temporal and spatial manners. Certain lncRNAs regulate the expression of genes related to the regulation of spermatogenesis (Figure 4). In mammals, lncRNAs are mainly involved in regulating spermatogenesis through cis or trans-actions on target genes. They also show different levels of expression and tissue specificities at different stages of spermatogenesis, thus indicating that they are involved in specific signaling pathways as guide molecules. LncRNAs regulate the SSCs, meiosis and spermatogenesis through controlling the transcriptional levels of downstream genes [35]. With regard to infertility treatment, the application of lncRNAs as transcriptional regulators of spermatogenesis has obvious advantages in that lncRNAs regulating does not involve protein translation. This strategy will also incur a more rapid response time during the regulation of spermatogenesis.

### 3.1. Regulation of LncRNAs in Spermatogonial Stem Cells

The process of spermatogenesis includes the proliferation and differentiation of SSCs, as well as the maturation of sperm. SSCs are male germline stem cells that support spermatogenesis and maintain male fertility [84,85]. The balance between the proliferation and differentiation of SSCs is regulated by the exogenous environment and endogenous genes. Recent research on lncRNAs confirmed that they are involved in regulating the proliferation and differentiation of SSCs through the regulation of endogenous gene expression. Previous functional annotation of lncRNA expression profiles in SSCs showed that differentially expressed lncRNAs participate in the control of SSC differentiation [86]. For example, during mouse spermatogenesis, Liang et al. [87] showed that 241 specific lncRNAs (including intergenic lncRNAs, antisense lncRNAs and sense overlap lncRNAs) could control SSCs survival, proliferation and differentiation via protein-coding genes and miRNAs. To further confirm the function of DELs, Liang et al. [88] found that the overexpression of lncRNA Gm2044, which was highly expressed in spermatogenesis, significantly inhibited male germ cell proliferation. LncRNA Gm2044 has also been shown to be abundant in mouse spermatocytes. It can suppress the translation of adjacent genes and inhibit spermatogonial proliferation by binding to an undifferentiated embryonic cell transcription factor [89]. Furthermore, the expression levels of lncRNA Gm2044 were shown to be significantly higher in non-obstructive azoospermia with spermatogonial arrest [88]. The potential function of lncRNAs to regulate the proliferation and differentiation of SSCs may help us to elucidate the regulatory mechanisms of SSCs, and further to develop novel therapies for the treatment of male infertility. Table 1 shows that lncRNAs serve as essential regulators of SSCs and could be used as potential specific biomarkers. For example, in mice, AK015322 has been shown to be highly expressed in SSCs and promotes the proliferation of SSCs in vitro. AK015322 was shown to antagonize the function of miR-19b-3 as a decoy. Then, it attenuates the repression of its endogenous target transcriptional factor, Ets-variant 5 (ETV5), a pivotal gene for SSCs self-renewal [90]. In addition, several lncRNAs can also mediate the expression levels of key genes related to the proliferation of SSCs. LncRNA033862 is known to be involved in the self-renewal process of SSCs in mice [91,92]. This is an antisense transcript of the GDNF receptor alpha1 (Gfra1) which is necessary for the proliferation and maintenance of SSCs [91]. LncRNA033862 can mediate spermatogenesis self-renewal by interacting with the Gfra1 chromatin to regulate Gfra1 expression levels [92]. Other studies in mice have shown that Mrhl lncRNA may mediate the process of meiosis and spermatogonia differentiation. Mrhl lncRNA lies on the 15th intron of the phkb gene [93] and is negatively correlated with the expression level of Sox8, which can regulate meiosis via the Wnt signal pathway [94].

### 3.2. Regulation of LncRNAs in Meiosis and Spermatogenesis

Spermatogenesis is essential for male reproduction, and is a complex process which has been well described in many studies on protein-coding genes, mRNA and small RNAs. While during the regulation of this complex physiological process, whether the initiation and maintenance of spermatocyte meiosis, or the maturation of sperm ultimately relies on the lncRNAs participation. Current studies show that more than sixlncRNAs are dynamically expressed during sperm meiosis. For example, in mouse testes, 1700108J01Rik and 1700101O22Rik, two testis-specific lncRNAs, are expressed in round spermatids at prophase and participate in post-transcriptional gene regulation [17]. In addition, Anguera et al. [97] demonstrated the importance of lncRNAs in meiosis and that lncRNA-Tsx is involved in regulating meiosis in spermatocytes. LncRNA-Tsx is situated on the X-inactivation locus and is expressed in germ cells at meiosis. The knockout of lncRNA-Tsx results in apoptosis in spermatocytes during pachytene, thus indicating that the lncRNA-Tsx gene is involved in spermatocyte meiosis and spermatogenesis [97]. The specific dynamic expression of lncRNAs may be related to sperm motility and spermatogenesis [22,98]. An increasing body of evidence now shows that lncRNAs play various roles in spermatogenesis. For example, in mice, lncRNA R53 is involved in regulating the metaphase of meiosis, and its overexpression is known to inhibit the subsequent progression of meiosis. It is possible that lncRNA R53 participates in the separation of homologous chromosome pairs during metaphase meiosis, or participates in the transcriptional regulation of genes related to spermatogenesis genes [99]. In mice, lncRNA 4930463O16Rik is known to be related to the protein expression of Topaz1, a germ cell-specific gene that is highly conserved in mammals. The deletion of TOPAZ1 disrupts the expression of lncRNA 4930463O16Rik and can lead to male infertility [34]. In addition, lncRNAs are highly associated with the maturational process in sperm. In human males, lncRNA NLC1-C is expressed in spermatogonia and early spermatocytes and is mainly expressed in the cytoplasm. Research has shown that the expression levels of NLC1-C are lower in the cytoplasm of sperm undergoing maturational arrest. In contrast, the expression levels of NLC1-C are up-regulated in the nucleus [92,96]. Linc00574 (also known as Lnc-TCTE3-1-2) is an lncRNA that transcribes from chr6q27 in the vicinity of TCTE3. Some studies found that TCTE3 is one of the factors involved in sperm motility defects. Furthermore, its expression is regulated by linc00574 through a negative self-regulating mechanism with the assistance of the REST as a transcription suppressor factor [100]. Unlike the patterns of protein-coding gene regulation, lncRNAs can regulate overall gene expression in a slightly regulated manner. The wide-ranging regulatory role of lncRNA during spermatogenesis suggests that it may regulate overall gene expression during spermatogenesis, thereby affecting male germ cell differentiation.

### 3.3. Interactions between LncRNAs, miRNA, and mRNAs during Spermatogenesis

LncRNAs can regulate gene expression at the pre-transcriptional, post-transcriptional, and translational levels [101,102]. Research has shown that lncRNAs act as “sponges” for miRNAs, thereby regulating the expression of target genes [103,104]. This molecular mechanism acts via a ceRNA process. Recent studies have shown that some lncRNAs also regulate spermatogenesis by adsorbing miRNAs [105,106,107]. In cases of human non-obstructive azoospermia, Zhou et al. [105] constructed a ceRNA regulatory network that featured lncRNAs, miRNAs, and mRNAs, to investigate the role and mechanism of lncRNAs in ceRNA. They found that the ceRNA regulatory network consisted of 21 nodes and 26 edges, comprising four lncRNAs, 13 miRNAs, and four mRNAs. Of the four lncRNAs, lncRNA ANXA2P3 were shown to bind to miR-613 and miR-206 to inhibit the expression of transketolase (TKT) mRNA. This plays an important role in glycolysis in the pentose phosphate cycle and is involved in cell growth and self-renewal. These results indicated that lncRNA ANXA2P3 plays an important role in non-obstructive azoospermia via the ceRNA regulatory network and could be utilized as a potential emerging biomarker for the treatment of non-obstructive azoospermia [105]. A previous study of human non-obstructive azoospermia identified a ceRNA regulatory network consisting of 1296 interaction pairs of lncRNAs, miRNAs, and mRNAs. The lncRNAs and mRNAs were positively correlated, while the lncRNAs and miRNAs were negatively correlated [33]. Functional experiments targeting this ceRNA regulatory network showed that LINC00467 was positively regulated with TDRD6 and LRGUK [33], both of which are key regulators of human spermatogenesis and maturation [106,107]. It indicates that LINC00467 is a promising biomarker for male infertility. However, the specific mode of action of LINC00467 in human spermatogenesis remains to be elucidated. In sheep testes, Zhang et al. [108] showed that high-grain feeding affected testicular growth at sexual maturity and that this process was mediated by lncRNAs, as determined by lncRNA-mRNA interaction network analysis. Several lncRNAs-miRNAs were shown to be involved in the regulation of spermatogenesis, including lncRNA LOC105607399 and LOC105610178. Moreover, another recent study revealed that lncRNAs and mRNAs were abnormally expressed in patients with oligozoospermia and the proportion of downregulated lncRNAs and mRNAs was less than that of upregulated ones [109].

## 4. LncRNAs Are Promising Biomarkers for Dysfunction in the Male Reproductive System

Spermatogenic arrest is one of the causes of dysfunction in the male reproductive system. During this process, many lncRNAs are involved in the disruption of differentiation of specific spermatogenic cell types, thus inhibiting the formation of spermatozoa. For example, the knockout of testis-specific lncRNA 1 (Tslrn1) in mice testes reduced sperm production, rendering the mice infertile. This demonstrated that the abnormal expression of lncRNAs can affect male reproduction [22]. In mice, the lncRNA-Tsx gene is expressed by germ and stem cells and is located proximal to the X-inactivation site. The knockdown of this gene results in smaller testes and abnormal X-inactivation sites in stem cells, although the resulting offspring are able to survive [97]. In humans, the knockdown of narcolepsy candidate 1 gene (NLC1-C) accelerates germ cell apoptosis, whereas its overexpression promotes germ cell proliferation [92]. Collectively, these studies indicate that specific lncRNAs are associated with spermatogenesis. 

### 4.1. LncRNAs in Livestock

In livestock, research on the proliferation of bovine male germ stem cells detected high expression levels of lncRNA H19 in bovine testicular tissue [110]. LncRNA H19 was one of the first imprinted genes to be identified and is expressed from the imprinted gene Igf2 [111]. This lncRNA was shown to be involved in cell proliferation and differentiation, as well as spermatogenesis [112,113,114]. In addition, H19 has been shown to modulate the IGF-1 signal pathway, which sustains the survival of a wide range of stem cells and is engaged in the proliferation and differentiation of male germinal stem cells [115,116]. Experimental reduction of the expression of lncRNA H19 significantly down-regulated the expression levels of IGF-1R in cattle mGSCs [110]. Another study showed that the expression of lncWNT3-IT could affect spermatogenesis in goats [36]. LncWNT3-IT is expressed in Sertoli cells and is related to the WNT3 protein which regulates proliferation and differentiation in testicular Sertoli cells [117,118]. In goat testes, the proliferation of Sertoli cells was shown to be upregulated with increased expression levels of WNT3 producing overexpression of lncWNT3-IT. In a previous study, He et al. [36] showed that lncWNT3-IT enhances WNT3 gene expression by clustering near to the WNT3 gene promoter and modulates the translation of WNT3 in a cis-acting manner, thereby regulating the growth cycle in goat Sertoli cells. Another lncRNA, lncNONO-AS, has been identified in goat testes. This lncRNA is mainly expressed in the nucleus and plays an important role at the epigenetic level. It also regulates spermatogenesis and testicular development by regulating the expression of the androgen receptor (AR) [119]. The overexpression of lncNONO-AS can increase the expression of AR by mediating NONO. In turn, this regulates spermatogenesis and participates in reproductive development in males. These studies show that lncRNAs play important roles in livestock spermatogenesis and provide us with a better understanding of the causes of infertility in mammals. Since lncRNAs are known to be involved in spermatogenesis and have been verified and localized, it follows that these may represent biomarkers for infertility in livestock. Further insights into the role of lncRNAs in livestock fertility has been acquired from sequencing studies in poultry. Comparisons of the histological characteristics and transcriptomics of two different breeds of goose identified a total of 462 differentially expressed mRNAs and 329 DELs (280 up-regulated and 49 down-regulated). These differentially expressed RNAs may be related to spermatogenesis [120]. In chickens, Liu et al. [121] investigated the lncRNAs associated with extreme sperm motility in rooster testes and identified a total of 2597 lncRNAs (including 1267 lincRNAs, 975 anti-sense lncRNAs, and 355 intronic lncRNAs), of which 124 were DELs. Several key lncRNAs have been shown to be involved in male germ cell differentiation, including XLOC_240662, XLOC_362463, and ALDBGALG0000002986 [86]. XLOC_240662 may modulate the progression from embryonic stem cells (ESCs) to SSCs by Sox9, which regulates the meiosis and differentiation of spermatogonia via the Wnt signal pathway [122,123]. XLOC_362463 can modulate the formation of SSCs via the TGF-β/BMP signal pathway. ALDBGALG0000002986 may be involved in autophagy during the formation of SSCs by targeting mTOR. This process is involved in the proliferation and differentiation of SSCs [124,125].

### 4.2. The Role of LncRNAs in Model Animals

A testis-specific lncRNA, Tesra, was identified during the meiotic phase of mouse spermatogenesis. This lncRNA is widely present in germ cells and the extracellular environment. Nuclear Tesra can prompt Prss42/Tessp-2 promoter activity and its binding to the promoter, and could therefore regulate the process of meiosis from secondary spermatocytes to round spermatids [126,127]. Another testicular-specific lncRNA that has been reported is lncRNA5512. Although knocking out lncRNA5512 did not affect spermatogenesis or fertility, this lncRNA is abundantly expressed in spermatocytes and round spermatids. Its specific localization in spermatocytes and round spermatids suggests that this lncRNA may be a useful biomarker for identifying spermatocytes and round sperm cells in mouse testes [128]. In mice, lncRNA-Tsx knockout results in the apoptosis of pachytene spermatocytes, thus indicating that the lncRNA-Tsx gene plays a role in both testicular development and other spermatocyte function [97]. Liang et al. showed that lncRNA Gm2044 can be used as a miR-335-3p sponge to increase the expression levels of the miR-335-3p target protein Sycp1 [129], which is specifically expressed in spermatocytes and regulates meiosis during spermatogenesis [130,131]. In addition, the up-regulation of lncRNA Gm2044 during spermatogenesis is regulated by A-MYB [129], a member of the Myb gene family [132]. Abnormal expression levels of A-MYB in mice are known to lead to male infertility [133]. In mice, the distal promoter region of lncRNA Gm2044 can bind with A-MYB. This complex can then promote the expression of lncRNA-Gm2044, thus, increasing the expression of Sycp1 [129].

### 4.3. LncRNAs in Humans

Research into human sperm has identified many lncRNAs related to spermatogenesis and male fertility, several of which are tissue-specific (Figure 5) [134]. For example, lnc98487, lnc09522, and lnc32058 were shown to be differentially expressed in dysfunctional sperm and normal sperm [134]. All of these lncRNAs were expressed in sperm from infertile men and were intercorrelated, thus suggesting that they may be associated with sperm viability. To better understand the reasons for spermatogenic failure in humans, Jan et al. [119] created a transcriptomic dataset of distinct and well-defined germ cell subtypes (deposited in NCBI’s Sequence Read Archive under accession number SRP069329). Using next-generation RNA sequencing, these authors also found that 137 lncRNAs and 110 RNA-binding proteins were significantly expressed in specific cells in human testis samples [135]. In another study of the expression of lncRNAs in human spermatogenesis, Rolland et al. [136] detected 1303 DELs in human and mouse testicular cells, including 113 lncRNAs that were dynamically transcribed during spermatogenesis. LncRNA ANO1-AS2 has also been shown to play an important role in spermatogenesis [137]. ANO1-AS2 is located close to the anoctamin1 (ANO1) gene which is a component of the transmembrane system and modifies gene expression in idiopathic infertile men [138]. The expression levels of ANO1-AS2 (linc02584) were negatively correlated with ANO1, and the expression levels of ANO1 were positively correlated with sperm motility and morphology. These results may be because lncRNA ANO1-AS2 is likely to downregulate the ANO1 gene by interacting with the ANO1 gene promoter, thus influencing sperm motility and morphology [137]. In summary, these studies provide new insights into the mechanisms of male infertility and suggest new biomarkers and therapeutic agents.

## 5. Intergenerational Inheritance of LncRNAs

Changing the environment and behavior of the parental generation can trigger epigenetic variation thus resulting in different environmentally adapted traits. Some of these traits are retained and passed on from parent to offspring by epigenetic regulation, thus resulting in changes in the expression of heritable genes without altering the gene sequence and intergenerational inheritance of the trait. Several studies have shown that ncRNAs play an essential part in intergenerational inheritance in this manner. For example, a high-fat diet in mice altered the abundance of sperm miRNAs in the sire, causing elevated levels of gene damage and reactive oxygen species levels in sperm. These metabolic disturbances were also observed in the offspring [139]. In addition to these effects of changes in parental dietary habits on the offspring, another study has shown that trauma can alter the expression of sncRNAs in the sperm of male mice, thus resulting in depressive behavior and altered glucose metabolism in the offspring [140]. Several other studies suggest that disrupted glucose metabolism can affect the quality of spermatozoa [141,142,143]. Other studies have shown that lncRNAs are involved in intergenerational genetics [144,145]. Yan et al. [146] showed that in mice, excessive NO_2_ inhalation in pregnancy results in neurological dysfunction in male offspring. It may be associated with the increased expression of lncRNA Malat1, and increased Malat1 perhaps then acts as a brain growth regulator by regulating the expression of *ApoE*. While *ApoE* plays an active role in maintaining brain development [144,147]. Furthermore, in mice, Li et al. [145] showed that a high-fat diet rich in n-3 polyunsaturated fatty acids can protect female offspring from the risk of mammary tumors. The mechanism of action was associated with the upregulation of lncRNAs in the p53 signaling pathway (upregulation of the p53 signaling pathway led to an increase in apoptosis and a decrease in cell proliferation) and a downregulation of lncRNA in the NF-κB and Jak-STAT signaling pathways (the downregulation of both signaling pathways led to a reduction in cellular estradiol). Paternal obesity has been shown to affect offspring metabolism the epigenetic reprogramming of spermatogonial stem cells; obesity may also lead to male sterility [148,149,150]. An et al. [151] showed that lncRNAs, such as Neat1 and Malat1, are involved in the intergenerational inheritance of obesity and obesity-induced reduced fertility. Furthermore, they serve as genetic vectors for the induction of paternal inheritance of obesity. Malat1 is located in the nucleosome and is involved in various biological processes [152]. In mice, Malat1 negatively regulates the expression of Neat1 which is also located in the nucleosome and is involved in spermatogenesis. The reduced expression of Neat1 is associated with reduced semen quality and reduced fertility [151,152]. In summary, the above lncRNAs are involved in the intergenerational inheritance of different traits and play an important role in regulating intergenerational inheritance. However, their specific mechanisms are not fully understood and there are still many lncRNAs that remain unexplored and need to be further investigated.

## 6. Conclusions and Future Perspectives

LncRNAs are involved in the regulation of many processes in male reproduction, particularly in the proliferation, differentiation, and meiosis of SSCs as they develop into mature spermatozoa. As research progresses, the functions of an increasing number of lncRNAs have been verified and shown to be advantageous in the regulation of male reproduction, including NLC1-C, Neat1, H19, lncNONO-AS, Mrhl, and lncRNA-Gm2044 (Table 2). Gaining a better understanding of lncRNAs holds tremendous promise for unravelling the regulatory mechanisms of spermatogenesis and male sterility and in identifying exciting new therapies because they are easily targeted by nucleic acid drugs [153]. The development of nucleic acid drugs has begun to provide successful solutions to male fertility problems and has allowed the development of drugs based on lncRNAs to combat male infertility [88]. In the future, traditional reproductive biology combined with high-throughput sequencing, bioinformatics, multi-omics, and other disciplines could deliver more potential regulatory targets for the treatment of male infertility.

## Figures and Tables

**Figure 1 ijms-22-13579-f001:**
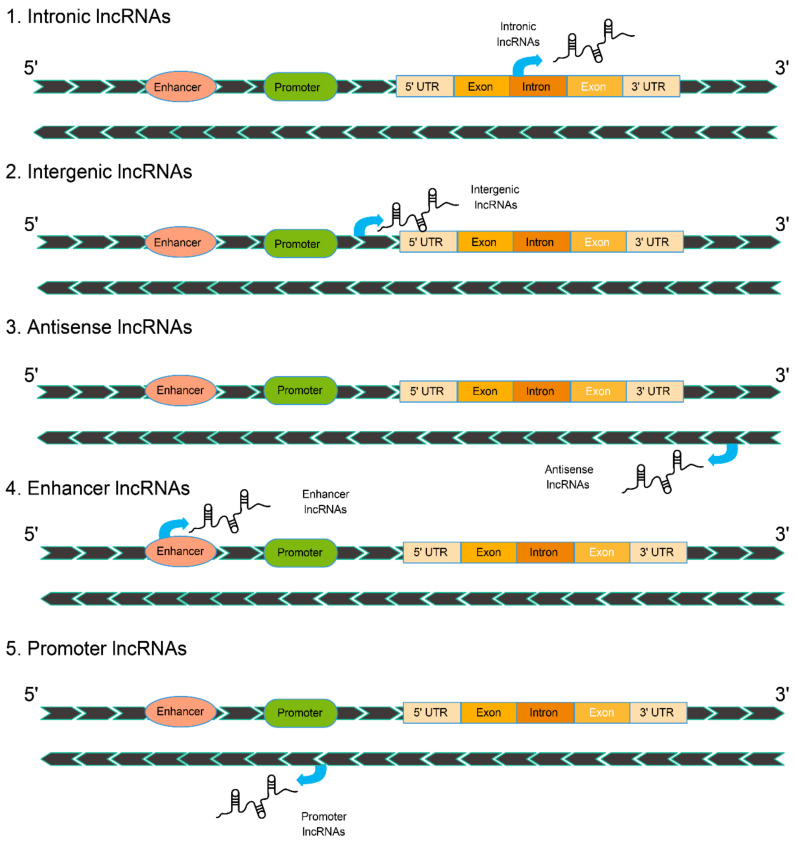
The biogenesis of long noncoding RNAs (lncRNAs) according to their locations and transcriptional origins.

**Figure 2 ijms-22-13579-f002:**
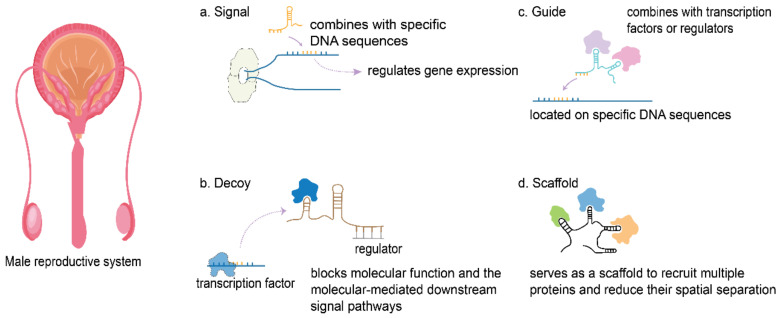
Classification of long noncoding RNAs (lncRNAs) according to their mode of action.

**Figure 3 ijms-22-13579-f003:**
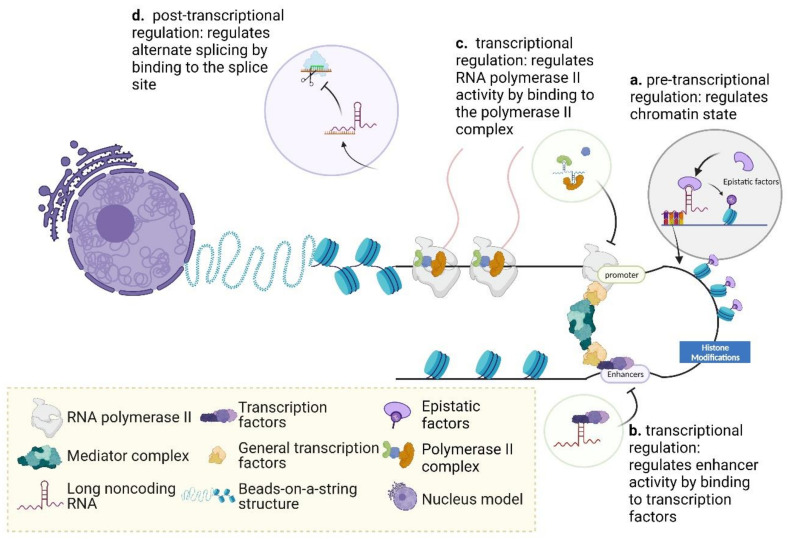
The regulatory mechanisms of Long Noncoding RNAs.

**Figure 4 ijms-22-13579-f004:**
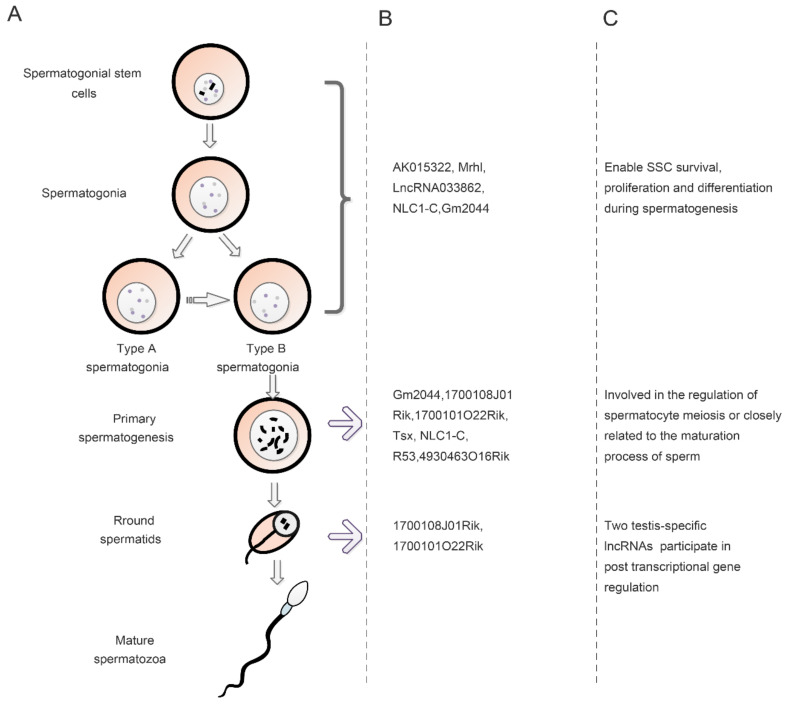
The different stages of spermatogenesis during which a spermatogonium undergoes division, differentiation, and meiosis to produce spermatozoa. (**A**) The process of spermatogenesis. (**B**) The lncRNAs that are involved in the regulation of spermatogenesis. (**C**) The primary function of lncRNAs in the regulation of spermatogenesis. Gonocytes produce spermatogonial stem cells in the postnatal testes; these develop from the primordial germ cells. Spermatogonial stem cells then undergo mitosis to produce type A spermatogonia, thus producing spermatogonial stem cells. These then develop into type B spermatogonia, which differentiate germ cells into diploid primary spermatocytes. During meiosis I, primary spermatocytes divide into secondary spermatocytes. During meiosis II, the secondary spermatocytes produce four haploid spermatids. During spermiogenesis, spermatids differentiate to produce spermatozoa through morphological changes, including flagellum development, nuclear condensation, acrosomal formation, and cytoplasmic reorganization.

**Figure 5 ijms-22-13579-f005:**
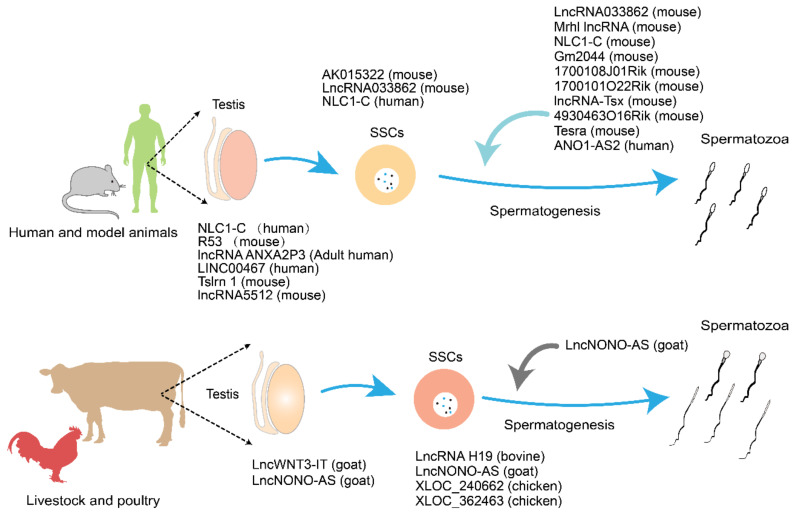
Long noncoding RNAs (lncRNAs) with possible potential as therapeutic biomarkers in mammals.

**Table 1 ijms-22-13579-t001:** Recent findings relating to the roles of long noncoding RNAs (lncRNAs) and their functions in spermatogonial stem cells (SSCs).

LncRNA Name	Location	Cell Type	Description [Ref]
AK015322	Chromosome 12, NC_000078.7	Mouse SSCs line C18-4	Highly expressed in spermatogonial stem cells; antagonizes the function of miR-19b-3 as a decoy; attenuates the repression of its endogenous target transcriptional factor, Ets-variant 5 (ETV5), which is a pivotal gene for SSC self-renewal [90].
LncRNA033862	Chromosome 19, NC_000085.7	Mouse SSCs and early spermatogonial cells	An antisense transcript of the GDNF receptor alpha1 (Gfra1); highly expressed in SSCs and early spermatogonia; regulates Gfra1 expression level through interactions with Gfra1 chromatin; and maintains the self-renewal and survival of SSCs [92].
Mrhl lncRNA	Chromosome 8	Mouse spermatogonial cells	Important for meiotic progression and differentiation; and negatively regulates Wnt signaling. During differentiation of spermatogonial cells, it binds to SOX8 which directly regulates the expression of premeiotic and meiotic markers [94,95].
NLC1-C	Chr 21 (NC_000021.9)	Human testicular spermatogonia and early spermatocytes	Knockout of the narcolepsy candidate 1 gene (NLC1-C) accelerates germ cell apoptosis, whereas its overexpression promotes germ cell proliferation [96].

**Table 2 ijms-22-13579-t002:** Recent findings relating to long noncoding RNAs (lncRNAs) and their potential as biomarkers and therapeutic targets.

LncRNA Name	Location	Cell Type	Description [Ref]
AK015322	Chr 12, NC_000078.7	Mouse spermatogonial stem cells (SSCs) line C18-4	Highly expressed in SSCs; antagonizes the function of miR-19b-3 as a decoy; attenuates the repression of its endogenous target transcriptional factor, Ets-variant 5 (ETV5), which is a pivotal gene for SSC self-renewal [90].
LncRNA033862	Chr 19, NC_000085.7	Mouse SSCs and early spermatogonial cells	Antisense transcript of the GDNF receptor alpha1 (Gfra1); highly expressed in SSCs and early spermatogonia; regulates Gfra1 expression level through interactions with Gfra1 chromatin; maintains SSC self-renewal and survival [92].
Mrhl lncRNA	Chr 8	Mouse spermatogonial cells	Important for meiotic progression and differentiation; negatively regulates Wnt signaling. During differentiation of spermatogonial cells, it binds to SOX8, which directly regulates the expression of premeiotic and meiotic markers [94,95].
NLC1-C	Chr 21 (NC_000021.9)	Human testicular spermatogonia and early spermatocytes	Knockout of the narcolepsy candidate 1 gene (NLC1-C) accelerates germ cell apoptosis, whereas its overexpression promotes germ cell proliferation [96].
Gm2044	Chr 7	Mouse spermatocytes	Highly expressed in spermatogenesis; overexpression of lncRNA Gm2044 inhibits cell proliferation, and can be used as a miR-335-3p sponge to increase the expression levels of miR-335-3p direct target protein, Sycp1, which is expressed specifically in spermatocytes and regulates meiosis during spermatogenesis [88,89,129].
1700108J01Rik	Chr 14 (NC_000080.7)	Mouse round spermatids	A mouse testis-specific lncRNA; only expressed in testicular germ cells at the pre-meiotic and round sperm cell stages; involved in post-transcriptional gene regulation [17].
1700101O22Rik	Chr 12 (NC_000078.7)	Mouse round spermatids	A mouse testis-specific lncRNA; only expressed in testicular germ cells at the pre-meiotic and round sperm cell stages; involved in post-transcriptional gene regulation [17].
lncRNA-Tsx	Chr X (NC_000086.8)	Spermatocytes	Located at the X-inactivation center and expressed in meiotic germ cells; knockout lncRNA-Tsx results in the apoptosis of pachytene spermatocytes [97].
R53	Chr 4 (NC_007090.3)	Mouse testis	Involved in regulation of the metaphase of meiosis; overexpression of lncRNA R53 inhibits the subsequent progress of meiosis [99].
4930463O16Rik	Chr 10 (NC_000076.7)	Mouse meiotic germ cells	Related to the protein expression of Topaz1, which is a germ cell-specific gene highly conserved in mammals; the absence of TOPAZ1 disturbs the expression of 4930463O16Rik, resulting in male infertility [34].
lncRNA ANXA2P3	Chromosome 10 (NC_000010.11)	Adult human testis	Binds with miR-613 and miR-206 to inhibit mRNA TKT expression [105].
LINC00467	Chr 1	Human testis	A promising biomarker for male infertility; can positively regulate TDRD6 and LRGUK, both of which are key regulators of human spermatogenesis and maturation [33].
Tslrn 1	Chr X	Mouse testis	One of the X-linked lncRNAs, testis-specific long noncoding RNA 1 (Tslrn 1), knocking out Tslrn1 shows that males with deleted Tslrn1 show normal fertility, and have significantly reduced sperm count [22].
lncRNA H19	Chromosome 11 (NC_000011.10)	Bovine testes	Regulates the IGF-1 signaling pathway, which maintains the survival of a variety of stem cells, and participates in the proliferation and differentiation of male germ stem cells [110,111,115].
lncWNT3-IT	Chr 17	Goat testes	Overexpression of lncWNT3-IT increases the expression of WNT3, and up-regulates the proliferation of Sertoli cells [36].
lncNONO-AS	Chr X (NC_000023.11)	Goat testis	Mainly expressed in the nucleus; regulates spermatogenesis and testicular development in goats by regulating androgen receptor (AR) expression [119].
Tesra	Chr 1	Mouse spermatogenesis	Nuclear Tesra regulates meiosis from secondary spermatocytes to round spermatids by increasing Prss42/Tessp-2 promoter activity through binding to the promoter [127].
lncRNA5512	-	Mouse testis	May be a useful biomarker for identifying spermatocytes and round sperm cells due to its specific location in these cells, although knockout does not affect spermatogenesis and fertility [128].
ANO1-AS2 (linc02584)	Chr 11	Human spermatogenesis	High expression of ANO1-AS2 down-regulates the ANO1 gene by interacting with ANO1 gene promoter, which influences sperm motility and morphology [137].
Neat1	Chr 11 (NC_000011.10)	Paternal line and offspring of mice	Located in the nucleosome; involved in spermatogenesis, and reduced expression of Neat1 which is associated with reduced semen quality and reduced fertility [151].
Linc00574	Chr 6 (NC_000006.12)	Human sperm	Transcribed from chr6q27 in the vicinity of TCTE3, regulated TCTE3 through a negative self-regulating mechanism [100,109].

## Data Availability

Not applicable.

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
