# Peer review of "Long Noncoding RNAs: Recent Insights into Their Role in Male Infertility and Their Potential as Biomarkers and Therapeutic Targets"

_ijms, 2021, doi:10.3390/ijms222413579_

Round 1

Reviewer 1 Report

In this review, Zhao et al discuss the role of long non-coding RNAs in male infertility and their use as biomarkers and therapeutic targets. This review covers many aspects of lncRNA and would be of interest to researchers working in the field of spermatogenesis. However, while going through the review I lost track. The main reason behind this was this review looks more like a summary of several research papers without any synthesis of ideas. Most of the sentences are written in past tense much similar to those written in the research thesis. I would like to see a better description of ideas synthesized from the previous research papers and a more organized presentation. I would also request authors to describe how lncRNAs work at the transcription level, post-transcription level, during alternate splicing, translation, posttranslational modifications of proteins, etc as separate points. This will give readers an idea of why understanding the role of lncRNAs in spermatogenesis is important. This way readers can’t lose track of what they are reading and trying to understand.

General points:

  1. Lines 85-87: Are they relevant here?
  2. Lines 100-101: Please edit to make the meaning clear.
  3. 3: What B and C are representing. Please describe that.

Author Response

Dear Editor and Reviewers,

Thank you very much for your critical review and comments on our work. This is the list of corrections of our manuscript entitled " Long Noncoding RNAs: Recent Insights into Their Role in Male Infertility and Their Potential as Biomarkers and Therapeutic Targets" submitted to IJMS. We have deal with all the comments of the reviewers. All the corrections have been marked up using the "Track Changes" function and high light in the manuscript. The whole manuscript has been carefully checked and improved. Please feel free to inform me if there are still some questions or comments.

With best regards,

Sincerely yours,

Shanjiang Zhao

Response to Reviewer 1 Comments

Thank you very much for your careful review on our manuscript. Your constructive suggestions gave us a plenty of valuable information on how to write a high-quality paper. The answers to the questions are listed one by one as follows, and the line number is record based on the all tags in the Microsoft word.

Point 1: In this review, Zhao et al discuss the role of long non-coding RNAs in male infertility and their use as biomarkers and therapeutic targets. This review covers many aspects of lncRNA and would be of interest to researchers working in the field of spermatogenesis. However, while going through the review I lost track. The main reason behind this was this review looks more like a summary of several research papers without any synthesis of ideas. Most of the sentences are written in past tense much similar to those written in the research thesis. I would like to see a better description of ideas synthesized from the previous research papers and a more organized presentation. I would also request authors to describe how lncRNAs work at the transcription level, post-transcription level, during alternate splicing, translation, posttranslational modifications of proteins, etc as separate points. This will give readers an idea of why understanding the role of lncRNAs in spermatogenesis is important. This way readers can’t lose track of what they are reading and trying to understand.

Response 1: We fully agree with you that this is very important for how to go about writing a quality review. Initially, we wanted to highlight the viability of each lncRNA as a potential biomarker (which is the main thrust of the article), so the text describes in as much detail as possible the source of the lncRNA, the transcription site and the impact caused by knockdown or overexpression. However, this also makes the description of each lncRNA seem like a short story. We have reflected on your suggestions, reworked the grammar of the article, especially the tense of sentences. What's more, we added as many synthesis ideas as possible such as lines 160-199, 252-254, 275-279, 309-313, 347-350, 499-503. Furthermore, we have added descriptions of the pre-transcriptional, transcriptional and post-transcriptional regulation of lncRNAs as separate points based on your suggestion (Lines 160-199). To make it easier for readers to understand, we have also created a figure (Line 201) of the regulatory mechanisms of lncRNAs. Thank you for your suggestion.

Point 2: Lines 85-87: Are they relevant here?

Response 2: I'm so sorry. This was a low-level mistake. We have overlooked this text when revising the article and we apologize for this. We have removed this text in response. Thank you for your advice.

Point 3: Lines 100-101: Please edit to make the meaning clear.

Response 3:Based on your suggestion, we have modified this sentence (New Lines 112-114). Thank you for your advice.

Point 4:What B and C are representing. Please describe that.

Response 4: In fig 4, B represents the lncRNAs that are involved in the regulation of spermatogenesis at this stage, and C represents the primary function of lncRNAs at this stage in the regulation of spermatogenesis. We did not describe the illustration well, which may have confused the reader. According to your suggestion, we have explained A, B and C separately and added them to the notes (New Lines 221-223). Thank you for your advice.

Reviewer 2 Report

The Authors present a paper reviewing knowledge about the Long Noncoding RNAs and its effects on male infertility. The review is generally well written and focuses on relevant and recent literature although not completely comprehensive in some points. The overall evaluation is positive, but some clarifications are needed before publication. T

  • Linc00574 (also known as Lnc-TCTE3-1-2) is an lncRNA that transcribes from chr6q27 in the vicinity of TCTE3. Some studies found that TCTE3 is one of the factors involved in sperm motility defects and its expression is regulated by linc00574 through a negative self-regulating mechanism with the assistance of the REST as a transcription suppressor factor (32749594). It could be a good idea to spend a few lines commenting this in the manuscript.
  • Moreover, another recent study revealed  that lncRNAs  and  mRNAs  were  abnormally  expressed  in  patients  with  oligozoospermia and the proportion of downregulated 
    lncRNAs and mRNAs was less than that of upregulated ones (34001678). I look forward to hearing the authors’ opinion.

Author Response

Dear Editor and Reviewers,

Thank you very much for your critical review and comments on our work. This is the list of corrections of our manuscript entitled " Long Noncoding RNAs: Recent Insights into Their Role in Male Infertility and Their Potential as Biomarkers and Therapeutic Targets" submitted to IJMS. We have deal with all the comments of the reviewers. All the corrections have been marked up using the "Track Changes" function and high light in the manuscript. The whole manuscript has been carefully checked and improved. Please feel free to inform me if there are still some questions or comments.

With best regards,

Sincerely yours,

Shanjiang Zhao

Response to Reviewer 2 Comments

Thank you very much for your kind words about our work and for your valuable suggestions on the manuscript, which are very important to us. We have carefully revised the article according to each of your suggestions. The answers to the questions are listed one by one as follows, and the line number is record based on the all tags in the Microsoft word.

Point 1: Linc00574 (also known as Lnc-TCTE3-1-2) is an lncRNA that transcribes from chr6q27 in the vicinity of TCTE3. Some studies found that TCTE3 is one of the factors involved in sperm motility defects and its expression is regulated by linc00574 through a negative self-regulating mechanism with the assistance of the REST as a transcription suppressor factor (32749594). It could be a good idea to spend a few lines commenting this in the manuscript.

Response 1: Thank you for your suggestion. We have added this text to the original text based on your suggestion (Lines 305-309).

Point 2: Moreover, another recent study revealed that lncRNAs and mRNAs were abnormally expressed in patients with oligozoospermia and the proportion of downregulated lncRNAs and mRNAs was less than that of upregulated ones (34001678). I look forward to hearing the authors’ opinion.

Response 2: This is an excellent reference and we thank you for your help. We have revised the article based on your suggestions (Lines 347-350).

Round 2

Reviewer 1 Report

Manuscript has significantly improved but still, there is a need to improve writing and figure formating such as:

  1. Avoid longer sentences throughout the manuscript.
  2. Avoid unnecessary use of punctuation marks.
  3. Figure quality is poor. High-resolution figures should be used.
  4. If text appears in the image of the figure (e.g., axis labels), use a sans serif font between 8 and 14 points.

Author Response

Dear Editor and Reviewers,

Thank you very much for your critical review and comments on our work. This is the list of corrections of our manuscript entitled " Long Noncoding RNAs: Recent Insights into Their Role in Male Infertility and Their Potential as Biomarkers and Therapeutic Targets" submitted to IJMS. We have deal with all the comments of the reviewers. All the corrections have been marked up using the "Track Changes" function and high light in the manuscript. The whole manuscript has been carefully checked and improved. Please feel free to inform me if there are still some questions or comments.

With best regards,

Sincerely yours,

Shanjiang Zhao

Response to Reviewer 1 Comments

Thank you very much for your kind words about our work and for your valuable suggestions on the manuscript, which are very important to us. We have carefully revised the article according to each of your suggestions. The answers to the questions are listed one by one as follows:

Point 1: Avoid longer sentences throughout the manuscript.

Response 1:Thank you for your advice. Many of the long sentences in the article are not easy to understand, which can be confusing for the reader. We have revised the article based on your suggestions,such as lines 37-39, 43-45, 52-55, 74-75, 84-88, 131-135, 201-208, 322-326, 402-411, 467-469.

Point 2: Avoid unnecessary use of punctuation marks.

Response 2:According to your suggestions, we have re-proofed the manuscript and removed unnecessary punctuation marks (Lines 19-20, 36-39, 48-49, 52-56, 94, 159, 168, 213, 246, 296, 318,346, 373, 375, 384, 394, etc.). Thank you for your advice.

Point 3: Figure quality is poor. High-resolution figures should be used.

Response 3:We fully agree with you suggestion. At first, we overlooked that word automatically compresses figures, which can degrade the quality of the figures. Now, we have corrected this problem and all figures in the manuscript are 300 DPI. Furthermore, we have submitted illustrator of all figures to meet the journal's requirements for figures. (Line 116, 150, 194, 217 and 448)

Point 4: If text appears in the image of the figure (e.g., axis labels), use a sans serif font between 8 and 14 points.

Response 4:We have modified the images of the figure based on your suggestions.  Thanks again for all your suggestions, which are very important for how to write a high-quality article. (Line 116, 150, 194, 217 and 448)